# Deciphering brain organoid heterogeneity by identifying key quality determinants
Tom Boerstler[1,15], Daniil Kachkin[1,15], Elizaveta Gerasimova[1,2], Naime Zagha[1], Federica Furlanetto [3], Negar Nayebzade[3], Luke Zappia [4,5], Michelle Boisvert[4,12,13,14], Michaela Farrell[1], Sonja Ploetz[6], Iryna Prots[2], Martin Regensburger [6,7], Claudia Günther[8,9], Juergen Winkler[6,7], Pooja Gupta [1,10], Fabian Theis[4,5,11], Marisa Karow[3], Sven Falk[3], Beate Winner [1,7] ✉ & Florian Krach [1] ✉

Brain organoids derived from human pluripotent stem cells (hPSCs) hold immense potential for modeling neurodevelopmental processes and disorders. However, their experimental variability and undefined organoid selection criteria for analysis hinder reproducibility. As part of the Bavarian ForInter consortium, we generated 72 brain organoids from distinct hPSC lines. We conducted a comprehensive analysis of their morphological and cellular characteristics at an early stage of their development. In our assessment, the Feret diameter emerged as a reliable, single parameter that characterizes brain organoid quality. Transcriptomic analysis of our organoid identified the abundance of unintended mesodermal differentiation as a major confounder of unguided brain organoid differentiation, correlating with Feret diameter. High-quality organoids consistently displayed a lower presence of mesenchymal cells. These findings provide a framework for enhancing brain organoid standardization and reproducibility, underscoring the need for morphological quality controls and considering the influence of mesenchymal cells on organoid-based modeling.

The generation of self-organizing 3D neural structures from human pluripotent stem cells (hPSC) was a milestone in stem cell research[1]. Brain organoids mimic aspects of human brain development, including cellular architecture and diversity. Major advantages of 3D models include their spatial organization and more physiological cell-cell communication compared to 2D cultures[2,3]. Recent discoveries in molecular and cellular anthropology and human brain evolution underscore their methodological power[4–8]. Additionally, the system provides a valuable platform for modeling a range of neurological disorders. These include rare developmental disorders, such as Optiz syndrome[9] or other microcephalies[10,11], neurodegenerative disorders like Alzheimer's disease[12,13], or amyotrophic lateral sclerosis (ALS)[14], and infectious diseases like Zika virus (ZIKV) infection[15,16].

Despite these remarkable advances, challenges related to robustness, accuracy, and reproducibility still exist. Hence, organoid models should be evaluated carefully[17]. Brain organoids generated from the same cell line, even under identical conditions, show variations in spatial organization and cellular diversity[18]. Publications often fail to provide clear criteria for selecting organoids for experiments. They typically do not describe if and how many organoids were discarded during the differentiation process[19–21]. This creates challenges in assessing the reliability of experimental results and introduces potential bias in the selection of organoids for disease modeling. While in principle, any hPSC line can be used to generate brain organoids, only a few specific cell lines have been used (e.g., hESC lines H9 (WA09), H1 (WA01), and WIBR1/2/3 and iPSC lines IMR90 and Kucg2)[22–24]. However,

[1]Department of Stem Cell Biology, University Hospital Erlangen, Friedrich-Alexander-University (FAU) Erlangen-Nürnberg, Erlangen, Germany. [2]Dental Clinic 1 — Department of Operative Dentistry and Periodontology, University Hospital Erlangen, FAU Erlangen-Nürnberg, Erlangen, Germany. [3]Institute of Biochemistry, FAU Erlangen-Nürnberg, Erlangen, Germany. [4]Institute of Computational Biology, Computational Health Center, Helmholtz Center Munich, Neuherberg, Germany. [5]Department of Mathematics, School of Computing, Information and Technology, Technical University of Munich, Munich, Germany. [6]Department of Molecular Neurology, FAU Erlangen-Nürnberg, Erlangen, Germany. [7]Center for Rare Diseases Erlangen (ZSEER), University Hospital Erlangen, FAU Erlangen-Nürnberg, Erlangen, Germany. [8]Department of Medicine 1, University Hospital Erlangen, University Hospital Erlangen, FAU Erlangen-Nürnberg, Erlangen, Germany. [9]Deutsches Zentrum Immuntherapie (DZI), University Hospital Erlangen, Erlangen, Germany. [10]Core Unit for Bioinformatics, Data Integration and Analysis (CUBiDA), University Hospital Erlangen, FAU Erlangen-Nürnberg, Erlangen, Germany. [11]School of Life Sciences, Technical University of Munich, Freising, Germany. [12]Present address: Broad Institute of MIT and Harvard, Cambridge, MA, USA. [13]Present address: Department of Cancer Biology, Dana Farber Cancer Institute, Boston, MA, USA. [14]Present address: Harvard Biological and Biomedical Sciences PhD Program, Harvard University, Cambridge, MA, USA. [15]These authors contributed equally: Tom Boerstler, Daniil Kachkin. ✉e-mail: beate.winner@fau.de; flo.krach@fau.de

international guidelines recommend using a diverse array of human stem cell lines to ensure the robustness and generalization of findings[25].

Here, we aim to understand early determinants of brain organoid quality by generating a set of brain organoids from 12 different hPSC lines using an adaptation of the original Lancaster protocol[19]. All cell lines were obtained from healthy donors. After 30 days of differentiation, significant variability was observed both within the brain organoids from different hPSC origins and among those derived from the same cell line. Through a systematic framework of analyzing morphological features and transcriptional signatures, we identified the organoid Feret diameter as a key determinant of organoid quality. Via computational analysis of transcriptomic data, we estimated the cellular composition of our organoids and found a correlation between organoid quality and the proportion of mesenchymal cells (MC). The proportion of MC was also positively correlated with the organoids' Feret diameter. These findings can serve as a valuable resource for researchers working with brain organoids to handle diversity and reduce variability.

## Results

We used a panel of 12 hPSC lines consisting of two embryonic stem cell (hESC) lines (H9 and HuES6) and ten iPSC lines, including both commercially available and in-house generated lines (Supplementary Table S1). All iPSC lines were analyzed for TRA-1-60 expression as a surrogate marker of pluripotency. All lines had greater than 90% TRA-1-60 positive cells. For brain organoid differentiation, we utilized an adapted version of the original unguided differentiation protocol developed by Lancaster and Knoblich (2014)[19].

### Cellular and structural composition of brain organoids

After 30 days of differentiation, we acquired a set of morphologically diverse organoids and characterized their cellular composition and architecture (Fig. 1A, B). We randomly selected six organoids from each line for immunostaining analysis. Cryosections of the organoids were stained with antibodies against the mature neuronal marker MAP2 and neural stem cell marker SOX2. This allowed us to verify the neural composition and distribution within the organoids and the formation of ventricular-like structures (VLS) (Fig. 1C). This analysis revealed considerable variability among the organoids: some organoids failed to form VLS while others developed multiple VLS populated with SOX2+ and surrounded with MAP2+ cells, indicating a high degree of active neurogenesis (Fig. 1D). We randomly selected three organoids from each line and quantified CNS-progenitor

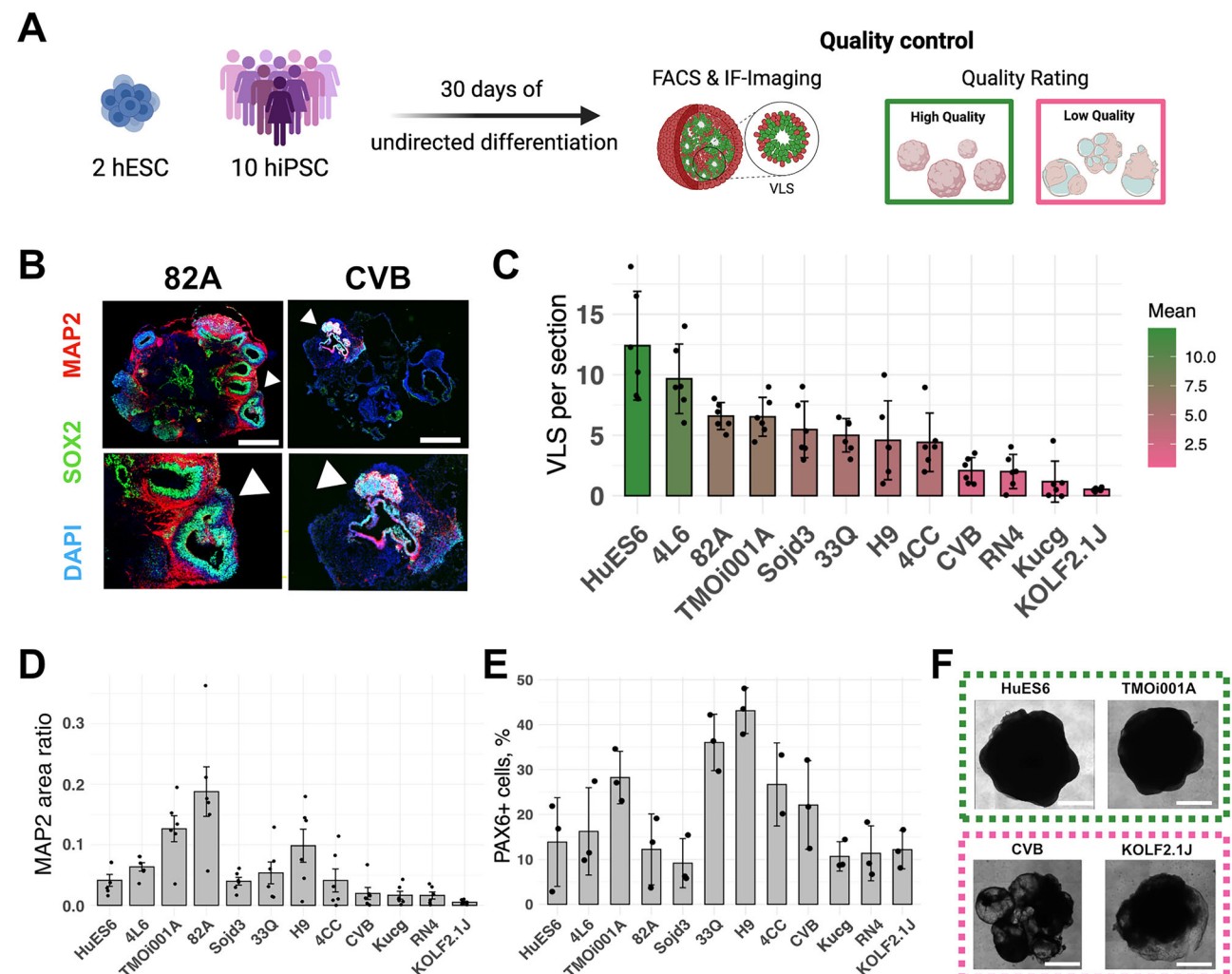

**Fig. 1 | Cellular and structural quality control of brain organoids. A** Experimental paradigm depicting undirected differentiation of 12 hPSC lines until day 30 of brain organoid, followed by standard quality control. **B** Immunofluorescence (IF) images of organoids stained at day 30 with DAPI (blue), for SOX2 (green), and MAP2 (red). Cropped regions below, highlighted with white arrows. White scale bar: 500 μm. **C** Barplot depicting the number of VLS per section. Each dot represents the amount of VLS per section of one organoid slice. The color scale depicts the increase in VLS, ranging from green (high amount) to pink (low mount). Data represented as mean ± SD (*n* = 6 organoids). **D** Barplot depicting the MAP2+ area ratio to whole organoid slice. Each dot represents one organoid. Data represented as mean ± SD (*n* = 6 organoids). **E** Barplot depicting the percentage of PAX6+ individual organoids assessed by flow cytometry. Data represented as mean ± SD. **F** Representative brightfield images of high and low quality rated organoids at day 30 of differentiation of the lines HuES6, TMOi001A, CVB, and KOLF2.1J. Scale bar: 1 mm.

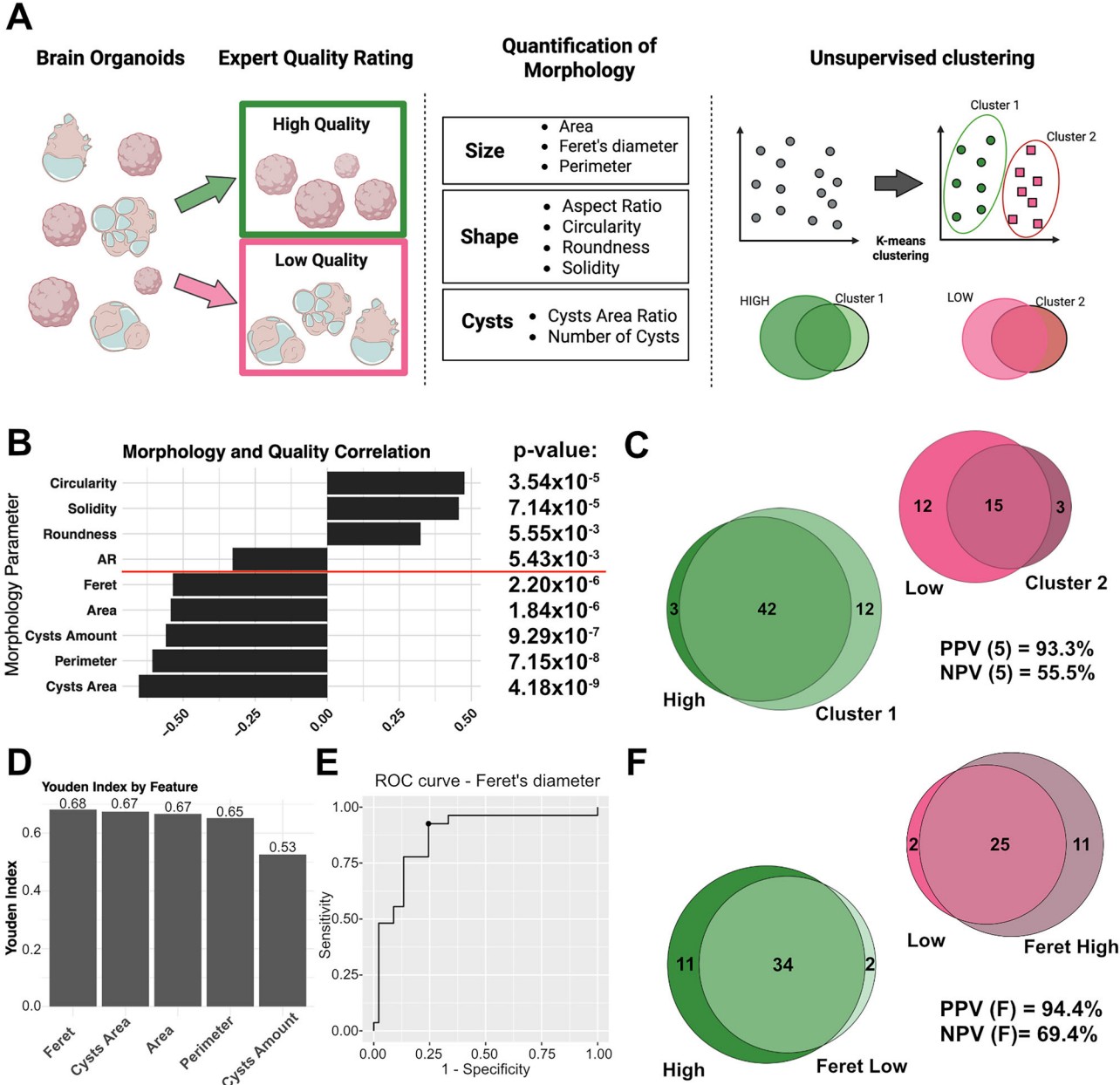

**Fig. 2 | Morphology assessment of brain organoids identifies Feret diameter as a classifier of organoid quality. A** Schematic illustrating the strategy to assess and determine morphological parameters that align with expert evaluation. **B** Bar graph depicting the result of the correlation analysis of morphological features with the expert's quality evaluation. The red square highlights five parameters surpassing the stringency threshold of $p$-value < $10^{-5}$. **C** Venn diagram depicting the overlap of organoids evaluated by an expert (green and pink, High, Low) or clustered in an unbiased way into two groups with k-means clustering using five highly correlated parameters (light-green and dark-pink, 'Cluster 1' and 'Cluster 2'). Positive (PPV) and negative prediction values (NPV) of the k-means clustering are depicted in the graph. **D** Bar graph illustrating Youden indices showing the diagnostic properties of five morphological parameters. The number on top of the graph represents the index value. **E** ROC curve for Feret diameter illustrating the ideal threshold with maximized specificity and sensitivity. Identified Feret diameter threshold: 3050 μm. **F** Venn diagram depicting the overlap of organoids evaluated by an expert (green and pink, High, Low) or clustered in two groups using Feret diameter threshold (light-green and dark-pink, 'Feret Low - Low Feret diameter 1' and 'Feret High - High Feret diameter 2'). Positive (PPV) and negative prediction values (NPV) of the k-means clustering are depicted in the graph.

marker PAX6 by flow cytometry (Fig. 1E). We observed a high degree of variability, both when comparing hPSC lines originating from different donors and among individual organoids originating from the same hPSC line (Fig. 1C–E). These findings highlight the heterogeneity in brain organoid development regarding VLS development and cellular composition.

## Morphology evaluation of brain organoids

To unbias the selection of brain organoids by expert evaluation, commonly used in the field, we developed a streamlined analysis approach.

Before, we randomly selected 72 individual organoids (six per line) for analysis. The generated organoids were classified as high-quality according to the critical morphological hallmark for brain organoids of a spherical shape interrupted by neuroepithelial buds growing into the Matrigel embedding (Fig. 1F, green). Low-quality organoids were classified by the presence of overt large fluid-filled cysts, overt migrating cells, or an irregular shape (Fig. 1F, red)[19,26]. We then captured brightfield images of the organoids and used ImageJ software to measure various morphological parameters. Subsequently, we applied a sequence of statistical

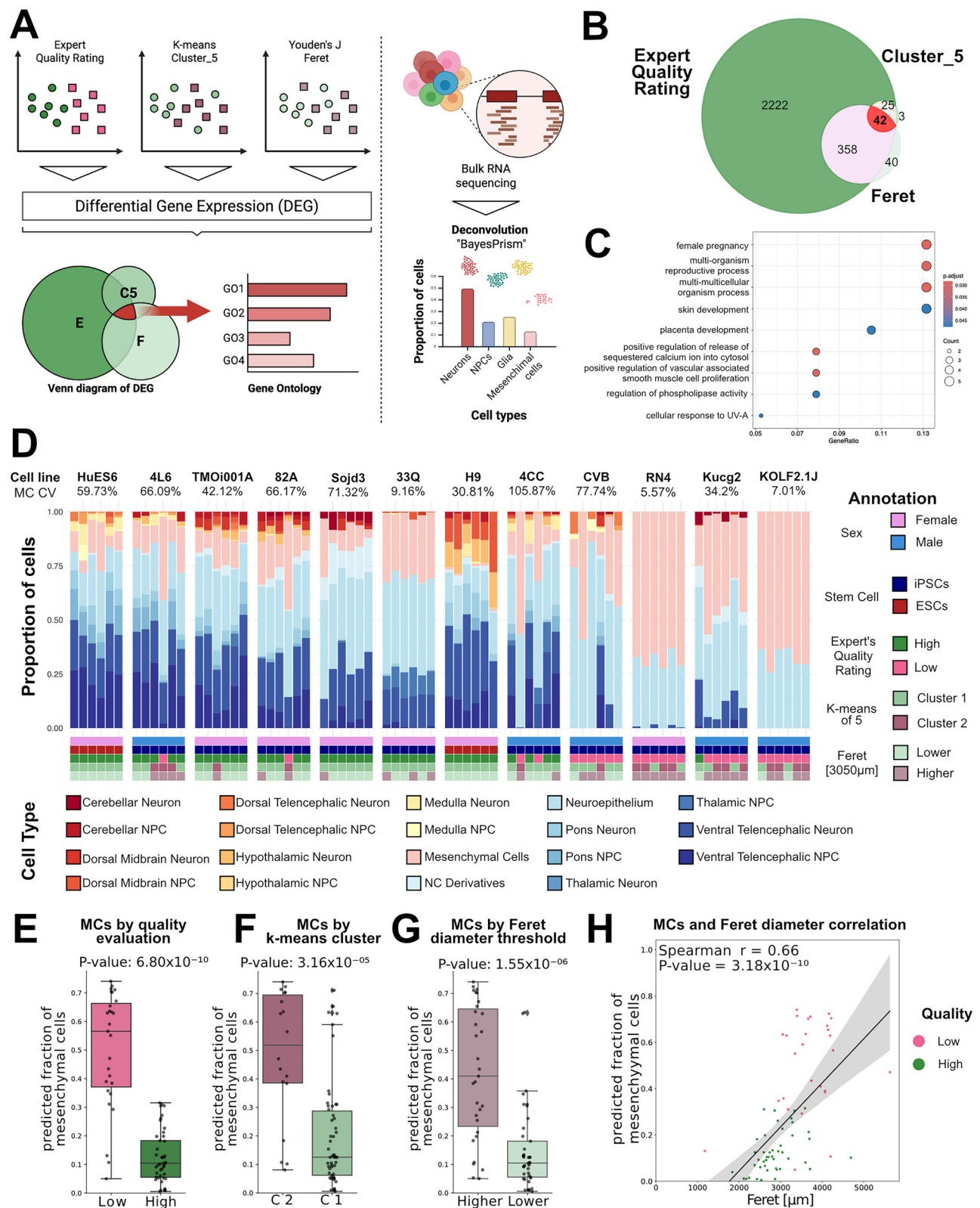

analyses to identify the morphological parameters that could predict the expert's evaluation (Fig. 2A).

To determine which distinct morphological parameters at day 30 are associated with the expert's evaluation, we conducted a point-biserial correlation analysis of the nine parameters (Supplementary Fig. 2B). We set an FDR-corrected *p*-value cutoff of $10^{-5}$ and a point-biserial correlation

coefficient |r-value| of >0.5 as predefined significance thresholds. Five morphological parameters satisfied this threshold: Feret diameter, Area, Cysts Amount, Perimeter, and Cysts Area (Fig. 2B).

To validate our findings, we asked whether these five parameters together can adequately reflect organoid quality in an unsupervised manner using k-means clustering. K = 2 was computed as the ideal

**Fig. 3 | Mesenchymal cells correlate with brain organoids morphology.**
**A** Illustration of the strategy for gene expression analysis. **B** Venn diagram, indicating the number of differentially expressed genes when organoids are compared by expert evaluation ('Expert's evaluation'), k-means clustering of five top correlating clusters ('Cluster_5') or the determined Feret diameter threshold ('Feret'). The overlap of all three analyses is depicted in red. **C** Gene Ontology analysis of differentially expressed genes common across three comparisons. The x-axis represents the Gene Ratio, while the y-axis displays the Gene Ontology (GO) terms associated with Biological Processes. The color scale indicates the adjusted *p*-value, and the size of the circles reflects the number of genes associated with each GO term. **D** Stacked bar chart illustrating the cellular composition of organoids calculated with deconvolution analysis. Further variables are color encoded below (sex, hPSC type, expert evaluation group, group in k-means clustering, group in Feret diameter

thresholding). Annotation coding for cell types is shown at the bottom of the graph. Each stack represents an individual organoid. The mesenchymal cell's coefficient of variation (MC CV) for each is presented as a percentage under the cell line name. **E** Boxplot of predicted fraction of mesenchymal cells in organoids by quality rating. Middle line represents median, individual measures as dots. Statistical significance shown on top (Wilcoxon rank-sum test). **F** Boxplot of predicted fraction of mesenchymal cells in organoids by cluster in k-means. Middle line represents median, individual measures as dots. Statistical significance shown on top (Wilcoxon rank-sum test). **G** Boxplot of predicted fraction of mesenchymal cells in organoids by Feret threshold of 3050 μm. Middle line represents median, individual measures as dots. Statistical significance shown on top (Wilcoxon rank-sum test). **H** Scatter plot of Mesenchymal cells (MC) and Feret diameter. Spearman r = 0.66, $p = 3.18 \times 10^{-10}$.

number of clusters for the given dataset (Supplementary Fig. S1). Interestingly, the clusters identified by k-means reflect the experts' evaluation to a high degree with a positive and negative predictive value (PPV and NPV) of 93.3% and 55.5%, respectively (Fig. 2C). This confirms that morphological measurements can objectify the visual expert evaluation. Next, we investigated if we could simplify this decision-making process to a single morphological parameter. We applied Youden's J statistics to the five morphological parameters to determine the threshold of a single parameter with the best diagnostic properties. All parameters computed thresholds with Youden indices above 0.5, indicating their positive prognostic properties (Fig. 2D and E). The Feret diameter (a maximal caliper diameter: the longest distance between any two points of the organoid) exhibited the best performance (Youden index of 0.68) at a threshold of 3050 μm (Supplementary Table S2, Supplementary Fig. S2). Strikingly, classifying the organoids by using this single parameter accurately reflected the expert's evaluation with a PPV and NPV of 94.4% and 69.4%, respectively (Fig. 2F). The similarity of these clustering methods to the expert's evaluation was also visually evident when putting the originally determined nine morphological parameters into principal component space (Supplementary Fig. S3A–C).

## Mesenchymal cell ratio correlates with the brain organoid morphology and quality

To uncover the cellular and molecular basis of organoid quality, we subjected the previously analyzed 72 organoids (Fig. 2) to bulk RNA sequencing for gene expression and cellular deconvolution analysis (Fig. 3A). First, we analyzed the dataset to identify genes that were differentially expressed when classifying the organoids by either the expert's evaluation, clusters obtained via k-means using five parameters (Fig. 2B) or only the Feret diameter threshold. Differential expression analysis, based on the expert evaluation, revealed the highest number of differentially expressed genes (2647). The other two classifying methods resulted in fewer differentially expressed genes (k-means: 70; Feret diameter: 441), but exhibited a large overlap with the expert's evaluation, validating our previous analyses. 42 genes were common to all classification methods (Fig. 3B). We performed Gene Ontology (GO) analysis on those 42 genes to uncover the underlying biology. The resulting GO analysis suggests the presence of cells not associated with neural lineage. (Fig. 3C). Therefore, we sought to estimate the cellular composition of our organoids and employed BayesPrism deconvolution analysis. BayesPrism is a computational tool that estimates the cellular composition of bulk RNA sequencing data by using a reference single-cell RNA sequencing dataset[27]. As a reference, we used a subset of the Human Neural Organoid Cell Atlas (HNOCA)[28]. The analysis revealed a heterogeneous, mostly neural cellular composition (range: 25.93–99.46%). Importantly, we observed a significant variation in the proportion of MC across the samples, ranging from 0.5% to 74% (Fig. 3D). The variability in MC composition among organoids derived from a single donor was lower (median coefficient of variation: 50.93%; range of coefficient of variation: 5.57–105.87%) than the variability observed when comparing the mean MC composition of organoids from different donors (coefficient of variation of mean MC composition in all cell lines: 80.98%). This suggests that MC

preferentially arise in specific hPSC lines while acknowledging a certain degree of heterogeneity when comparing organoids of the same donor. A significant difference in the fraction of predicted MC content was evident when sampling the organoids by expert quality rating, k-means cluster and Feret diameter (Fig. 3E–G). The significant positive correlation between Feret diameter and predicted MC content further illustrates that high-quality organoids are smaller and exhibit a lower content of MC (Fig. 3H). We validated this finding computationally by applying WebCSEA - Web-based Cell-type Specific Enrichment Analysis of Genes[29] as a second prediction approach with an alternative underlying statistical method as BayesPrism (t statistics vs. Bayesian approach). Using the 42 commonly differentially expressed genes (see source data Tab 3B), we found an association of our set of genes with non-neural cells like epithelial and stromal cells (fibroblasts and mesenchymal stem cells) in the dataset (Supplementary Fig. S4). At last, we aimed to validate our findings in all 12 hPSC lines. First, we selected two independent markers of MC, CD73 and CD105. These markers showed a significant positive correlation of their RNA expression with the Feret diameter and predicted MC fraction (Supplementary Fig. S5). We stained organoids from all 12 hPSC lines with both markers (Fig. 4A). Fluorescent intensities of both markers correlated well with each other in individual organoids confirming the validity (Fig. 4B). To explore whether the Feret diameter correlates with the expression of the two markers, we measured the section's Feret diameter and correlated it with the fluorescent intensities of the two MC markers. We identified a significant positive correlation for CD105 and CD73 with the section Feret diameter (Fig. 4C, D), validating our previous results. As we identified, the quality of organoids largely depends on the hPSC line, we separated the lines by their majority quality rating and confirmed that hPSC lines that produce rather higher quality organoids exhibit a reduced fluorescent intensity of CD105 and CD73 (Supplementary Fig. S6). Taken together, our experimental validation confirms that a larger Feret diameter and lower organoid quality are associated with a higher MC composition in organoids.

These data indicate that the quality of brain organoids can be evaluated using distinct morphological parameters. Notably, these parameters correlate with the presence of MC, as higher-quality organoids tend to contain fewer MC.

## Discussion

Here, we describe the heterogeneity of unguided brain organoids differentiated from multiple hPSC lines. Using a streamlined data analysis approach, we identify morphological parameters, most prominently the Feret diameter, as a simple-to-estimate surrogate of brain organoid quality. Through bioinformatic deconvolution analysis, we uncover that low-quality brain organoids contain a higher proportion of MC, accompanied by a high Feret diameter.

Hence, our results are an important contribution to the field's efforts to standardize brain organoid research.

We identified easily measurable morphological features that can be used to objectively assess organoid quality at the early stages of the differentiation process. Among those, the Feret diameter reflected the expert evaluation most precisely. A low Feret diameter indicates a higher brain

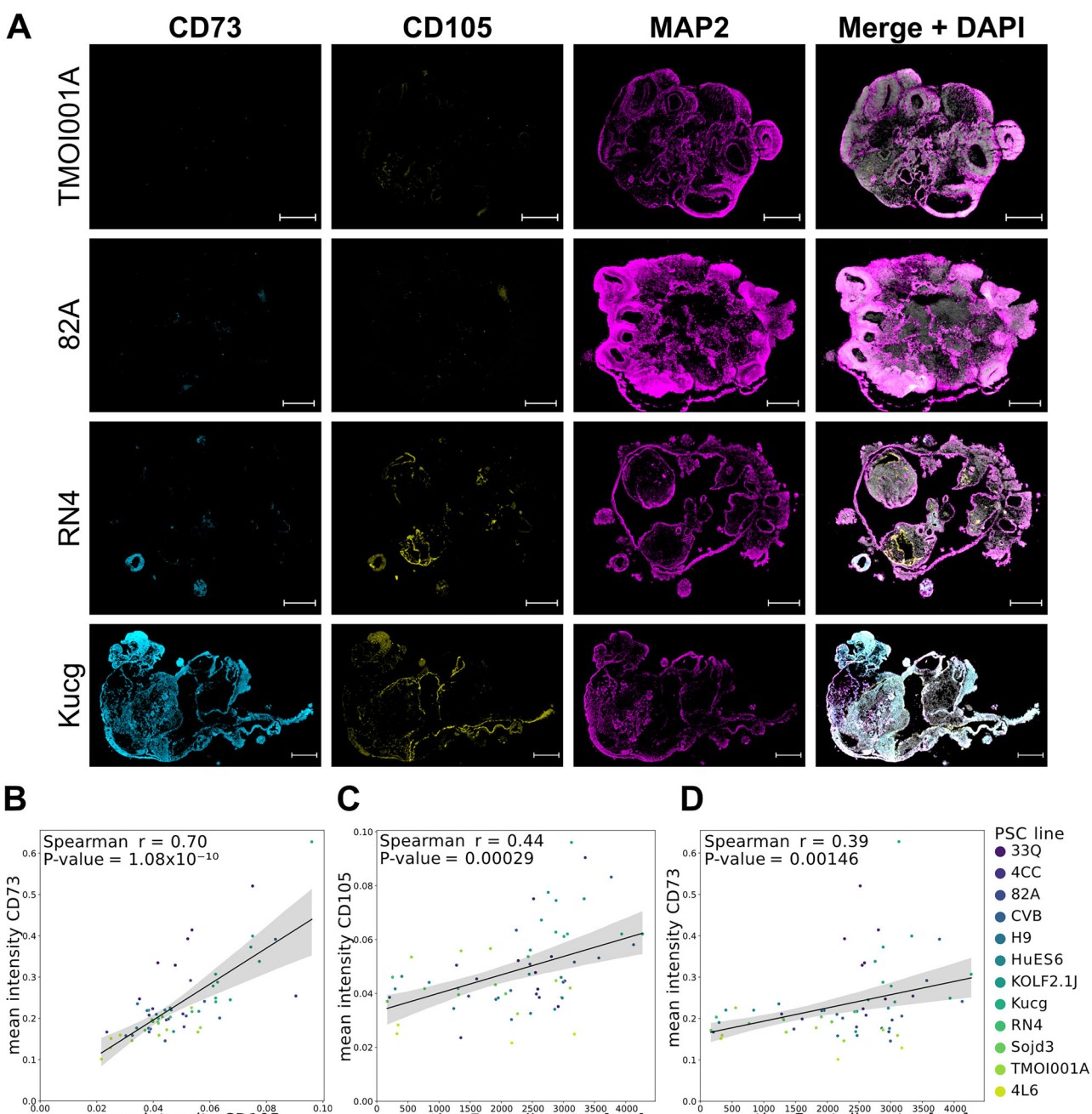

**Fig. 4 | Experimental validation illustrating that mesenchymal cells are associated with higher organoid Feret diameter in sections. A** Representative images of organoids from four hPSC lines. Stained with CD73 (blue), CD105 (yellow), MAP2 (magenta), and DAPI. Scale bar 500 μm. **B** Scatter plot depicting mean fluorescent intensity of CD105 (x-axis) and CD73 (y-axis). Individual dots represent individual organoids ($n = 65$). Color depicts 12 hPSC lines (legend on right). Significant positive correlation (Spearman r = 0.7, $p = 1.08 \times 10^{-10}$) is depicted in the graph. **C** Scatter plot depicting the organoid section's Feret diameter (x-axis) and mean fluorescent intensity of CD105 (y-axis). Individual dots represent individual organoids ($n = 65$). Color depicts 12 hPSC lines (legend on right). Significant positive correlation (Spearman r = 0.44, $p = 0.00029$) is depicted in the graph. **D** Scatter plot depicting the organoid section's Feret diameter (x-axis) and mean fluorescent intensity of CD73 (y-axis). Individual dots represent individual organoids ($n = 65$). Color depicts 12 hPSC lines (legend on right). Significant positive correlation (Spearman r = 0.39, $p = 0.00146$) is depicted in the graph.

organoid quality. The area and number of cysts have been associated with low brain organoid quality in earlier studies[19,30]. However, their identification may require a certain experience in stem cell research. Brain organoids' asymmetrical increase in volume due to the number and size of cysts can be reflected by an increasing Feret diameter. Implementing this measurement as a parameter for organoid selection is consequent, as it is a straightforward measurement that can be performed using standard cell culture microscopes. It can be applied by investigators with minimal experience or may be automated. Hence, this finding could aid the growing scientific community that uses brain organoids but lacks extensive experimental expertise in the field.

We aimed to uncover the cellular composition of organoid quality by applying state-of-the-art deconvolution analysis of bulk RNA-seq from individual organoids. We found that MC are the main confounders in unguided cerebral organoid studies in low-quality organoids. The presence of MC in brain organoids derived from hPSCs has been reported before[5,28,31],

but not further characterized. Mesenchymal stem cells are frequently discussed to have neuroprotective properties in neurological disorders via cell-cell contact or secreted factors, and exosomes[32–34]. This cell type was also under clinical investigation in neurodegenerative disorders, e.g., Alzheimer's disease (Clinical Trial NCT02833792) and amyotrophic lateral sclerosis (Clinical Trial NCT03280056). However, no convincing proof of clinical efficacy leading to FDA or EU approval has been shown so far. In the context of CNS malignancies, the abundance of MC in tumor tissue inversely correlates with patient survival, and MC are thought to promote the proliferation of glioma cancer cells[35]. Hence, the presence of this cell type should caution researchers in disease modeling and preclinical development, as it may significantly confound results. Therefore, pre-screening brain organoids using a defined morphological parameter to obtain organoids with fewer cells of confounding potential can be an asset in therapeutic development.

Reduction of MC content with the addition of small molecules would be the consequential step to improve reliability. Guided brain organoid differentiation protocols rely on dual Smad inhibition, which can suppress mesodermal lineages[36–38]. However, recent publications have demonstrated the importance of extracellular matrix embedding in brain organoid morphogenesis, which is missing in most of the guided brain organoid differentiation protocols[39]. Our findings describe important characteristics of the unguided minimum morphogen differentiation that need to be considered in experimental planning. The reported variability between hPSC lines or even clones of organoids from healthy individuals underscores the need for standardized and unbiased organoid selection criteria, especially when investigating phenotypes between healthy and diseased conditions.

In line with stem cell community's efforts to publish international guidelines, our findings provide insights into the cellular confounders of brain organoid heterogeneity and propose an easy-to-use framework for researchers to increase the experimental reliability[40].

## Methods
### Human pluripotent stem cell line maintenance and processing
We used a total of 10 previously published iPSC lines derived from healthy human tissues—UKERi4L6, UKERi4CC, UKERiRN4, UKERi33Q, UKERi82A, TMOi001A, KOLF2.1, Kucg2, Sojd3, CVB, and two human ESC lines—H9 and HuES6 (Table S1).

Cell lines UKERi4L6, UKERi4CC, UKERiRN4, UKERi33Q, and UKERi82A[41,42] were generated at the University Hospital Erlangen (Erlangen, Germany). TMOi001-A line was bought from Thermo Fisher (Waltham, MA, USA). Kucg2 and Sojd3 lines were purchased from Wellcome Sanger Institute (WTSI) via the European Collection of Authenticated Cell Cultures (ECACC) (Porton Down, UK). The KOLF2.1 cell line[43] was obtained from the Jackson Laboratory (Bar Harbor, ME, USA). The CVB cell line was bought from the Coriell Institute for Medical Research (Camden, NJ, USA). HuES6 cell line was received from Harvard Stem Cell Institute, and H9 was obtained from WiCell Research Institute (Madison, WI, USA). IPSC lines UKERi4L6, UKERi4CC, UKERiRN4, UKERi33Q, and UKERi82A were generated at the Department of Stem Cell Biology. For iPSC generation, skin biopsies of study participants were obtained. iPSCs were generated from fibroblasts using the CytoTune iPS 2.0 Sendai Reprogramming Kit (Thermo Fisher Scientific, USA) according to the manufacturer's instructions. All cell lines have been operated in sterile conditions and tested negative for mycoplasma before being included in the experimental procedures. Cells were cultured in human stem cell media mTESR Plus (StemCell Technologies) with mTESR Plus supplements (StemCell Technologies). The medium was changed every other day. Cells were propagated with ReLeSR™ Passaging Reagent (StemCell Technologies, Canada) on 4 mg/ml Geltrex (Gibco, USA) coated polystyrene cell culture plates for growth. Low-pass whole-genome sequencing at a depth of 1.7x was performed for all cell lines using a NovaSeq 6000 (Illumina, USA). Reads were aligned to the human genome reference (hg19), and copy number variation (CNV) analysis was conducted using the DRAGEN CNV pipeline (ver. 3.10).

### Human brain organoid differentiation
To generate brain organoids, iPSCs and ESCs were differentiated using a protocol from Lancaster et al.[19] with slight adaptations. Briefly, prior to brain organoid differentiation, cells were treated with Accutase (Gibco, USA) for 5 min at 37 °C to dissociate cells and have a single-cell suspension. The cells were resuspended in Organoid Formation Media (OFM) (DMEM/F12 with GlutaMAX, 20% Knockout Serum Replacement, 3% FCS, 1% MEM-NEAA, 50 μM β-Mercaptoethanol (all Gibco, USA) with 50 μM ROCK inhibitor and 4 ng/ml FGF2 (PeproTech, USA). Embryoid bodies (EBs) were generated by adding 10,000 cells per well of a 96 U-bottomed plate, which were then centrifuged at $600 \times g$ for 5 min and incubated in OFM supplemented with ROCK inhibitor (Enzo Life Sciences, Germany) and FGF2 (PeproTech, USA). On day 3, two-thirds of the media were replaced with fresh OFM without ROCK inhibitor and FGF2 supplements. On day 5, the media was completely changed from OFM to NIM (Neural Induction Media) consisting of DMEM/F12 with GlutaMAX, 1% N2, 1% MEM-NEAA, and 1 μg/mL Heparin. The media was changed every other day. On day 11, each organoid was embedded into a Matrigel (Corning, USA) droplet and placed into a 6-well plate with Cerebral Differentiation Media (CDM) (50% DMEM/F12 (Gibco, USA), 50% Neurobasal (Gibco, USA), 0.5x N2 (17502048, Gibco), 1x B27 without Vitamin A (12587-010, Invitrogen), 1x Pen/Strep (15140-122, Gibco), 0.5x MEM-NEAA (11140-35, Gibco), 0.025% Insulin (I9278, Sigma), 50 μM β-Mercaptoethanol (Carl Roth, Germany)). The organoids were incubated without agitation until day 15 and fed every other day with CDM. On day 15, the CDM was changed by supplementing B27 with Vitamin A (17504-044, Invitrogen) instead of B27 without Vitamin A. The 6-well plate was then transferred to an orbital shaker (Laborschüttler Rocker 3D basic, IKA, Germany), rotating at 33 rpm. The media was changed every other day until day 30. On day 30 of differentiation, brain organoids with distinct morphology were chosen for further gene expression analysis using RNA sequencing.

### Brightfield imaging and morphometry analysis
To monitor the growth dynamics of brain organoids, random organoids from each cell line were selected and imaged every five days from day 11 to day 30 using a Zeiss Axio Vert.A1 microscope. For larger organoids, multiple overlapping images were captured and stitched together to create a single composite image for comprehensive visualization. On day 30, a subset of organoids (six per cell line) was selected for RNA sequencing, and these organoids were imaged using a Zeiss Stemi 2000-CS microscope.

For morphological analysis, images of the organoids were processed using ImageJ software (version 2.14.0). Prior to analysis, all images were converted to 8-bit grayscale format to standardize the processing and ensure compatibility with thresholding functions. Automatic thresholding was applied to distinguish organoid structures from the background, followed by manual adjustments to optimize segmentation and accurately define the organoid boundaries. This ensured that the entire structure of each organoid was captured for analysis.

Subsequent morphological measurements were conducted. Area, Min and Max gray values, Shape descriptors, Mean gray value, Feret diameter, Median, and Kurtosis were initially chosen for morphological characteristics of brain organoid images using the Set Measurements function in ImageJ. Later, the following parameters were selected for a better description of the shape of the organoids: Area, Perimeter, Aspect Ratio, Feret Diameter, Roundness, Circularity, and Solidity. These metrics provided quantitative data on organoid size, shape, and structural integrity. Area and perimeter measurements were used to assess the organoid size, while shape descriptors such as aspect ratio, circularity, and roundness offered insights into the structural form of the organoids. Solidity was particularly useful in evaluating the compactness and overall structural integrity of each organoid.

On day 30, cystic structures within the organoids were also evaluated. Cysts were identified based on their morphology and quantified by counting the number of cysts per organoid. Additionally, the total cyst area was measured and expressed as a proportion of the entire organoid area to assess the prevalence of cyst formation across different cell lines. These data, along

with other morphological measurements, are presented in the supplemental source data table.

## Flow cytometry (FC)

The pluripotency of hPSCs was evaluated by flow cytometry using the TRA-1-60 marker (Biotec). To assess cellular composition, we quantified PAX6 and MAP2 cells in brain organoids. Brain organoids were dissociated by recovering them from Matrigel using a Cell Recovery Solution (Corning) and then using a papain-based Neural Tissue Dissociation Kit (Miltenyi Biotec) according to the manufacturer's instructions. Live-dead staining was performed with the LIVE/DEAD™ Fixable Dead Cell Stain Kit (Thermo Fisher Scientific) as per protocol. For intracellular staining, $5 \times 10^5$ cells were fixed with 4% paraformaldehyde for 15 min at room temperature. Cells were permeabilized and blocked with 2% FCS and Fc receptor block (BioLegend). They were then incubated with anti-PAX6-APC (1:50, Miltenyi Biotec, 130-123-267) and anti-MAP2-PE (1:100, Merck, FCMAB318PE) antibodies for 15 min at 4 °C in the dark. Flow cytometry was conducted using a CytoFLEX flow cytometer and analyzed with CytExpert software version 2.4.0.28 (both from Beckman Coulter). Fluorescence compensation was performed using UltraComp eBeads™ Compensation Beads (Invitrogen). For flow cytometry analysis, cells from the samples underwent gating for single cells, followed by gating for viable cells. They were then analyzed for MAP2 and PAX6 positivity. Unstained and fluorescence-minus-one samples were used as controls to establish gating parameters. Three brain organoids per cell line were analyzed for cellular composition via flow cytometry.

## Immunohistochemistry (IHC)

On day 30, organoids were washed twice with PBS and fixed in 4% paraformaldehyde (PFA, Carl Roth) for 1 h. Post-fixation, they were washed thrice with PBS (10 min each) and immersed in 30% sucrose for cryoprotection. Organoids were then embedded in Neg-50™ Frozen Section Medium (Thermo Fisher) on dry ice and stored at −20 °C. Organoids were cryosectioned into 30 μm slices using a CryoStar NX70 cryostat (Thermo Fisher) and placed on SuperFrost Plus™ slides (Thermo Fisher). Prior to antigen retrieval, organoid sections were fixed again in 4% PFA for 12 min, followed by two washes in PBS. For antigen retrieval, sections were incubated at 70 °C for 20 min in HistoVT One buffer (1:10 dilution, Thermo Fisher). Sections were blocked for 15 min with PBS containing 4% normal donkey serum (NDS, Sigma-Aldrich) and 0.25% Triton X-100 (Sigma-Aldrich), followed by a 2-h incubation at room temperature with the same blocking buffer. Primary antibodies diluted in antibody solution (PBS, 4% NDS, 0.1% Triton X-100) were applied overnight at 4 °C (SOX2: 1:300 (3579S, Cell Signaling Technology); MAP2: 1:500 (M9942, Sigma-Aldrich); COL1A1 (3G3): 1:100 (sc-293182, Santa Cruz)), CD73 (ab317364, abcam), CD105 (ab252345, abcam). The next day, sections were washed twice with PBS (5 min each) and once with PBS containing 0.5% Triton X-100 (8 min). Secondary antibodies and DAPI (1:1000, Sigma-Aldrich) were applied for 2 h at room temperature. Sections were washed thrice with PBS (5 min each) and mounted using Aqua Polymount (Polysciences). Immunofluorescence images were captured using an EVOS M7000 Imaging System (Thermo Fisher) for SOX2 and MAP2 staining and a Zeiss Observer Z1 fluorescence microscope (Zeiss) for CD73, CD105, and MAP2 staining. To evaluate the signal intensity from the COL1A1 antibodies, we measured the Corrected Fluorescent Intensity $CFI = Integrated\ Density - (Area \times Background\ Mean\ Intensity))$, using the ImageJ software version 2.14.0 and normalized it to the size of the organoid. Analysis of CD73, CD105 MFI and MAP2 area was performed with a customized CellProfiler 4 pipeline[44]. In the first step, "Smooth → Gaussian Filter" for DAPI images, followed by "IdentifyPrimaryObjects" and "SplitOrMergeObjects- Merging method Distance" to generate a single object for the whole organoid area. The MFI was assessed with "MeasuredImageIntensity" module for CD73, CD105, and MAP2. The Feret diameter per slice was evaluated with "MeasureObjectSizeandShape" module. Finally, the data was exported with "ExportToSpreadsheet" module.

## Total RNA isolation, quality control, library preparation, and sequencing

RNA was extracted using the RNeasy kit (Qiagen, Germany) according to the manufacturer's instructions. RNA concentrations were measured using a NanoDrop NP80 (Implen, Germany). A total of 1000 ng per sample was sent for RNA sequencing to Azenta Life Sciences (Genewiz Leipzig, Germany) for sequencing library preparation and 150 bp paired-end sequencing with Poly-A selection. 72 samples were sent and sequenced at a depth of >20 million reads in each sample. RNA samples were quantified using a Qubit 4.0 Fluorometer (Life Technologies, Carlsbad, USA), and RNA integrity was checked with an RNA Kit on an Agilent 5300 Fragment Analyzer (Agilent Technologies, Palo Alto, USA). The ERCC RNA Spike-In Mix kit (Thermo Fisher Scientific, USA) was added to normalized total RNA prior to library preparation following the manufacturer's protocol. RNA sequencing libraries were prepared using the NEBNext Ultra II RNA Library Prep Kit for Illumina following the manufacturer's instructions (New England Biolabs, USA). mRNAs were first enriched with Oligo(dT) beads. The samples were sequenced using a 2x150 Pair-End configuration (ver. 1.5) on an Illumina NovaSeq 6000. Image analysis and base calling were conducted by the NovaSeq Control Software (ver. 1.7). Raw sequence data was converted into .fastq files using the bcl2fastq program version 2.20.

## RNA seq analysis

To assess which gene expression affects organoid morphology and quality, we performed RNA sequencing of 72 organoids derived from 12 different stem cell lines. FastQC was used for the quality control of raw sequences (Andrews, S. (2010). FastQC: A Quality Control Tool for High Throughput Sequence Data. Available online at: http://www.bioinformatics.babraham.ac.uk/projects/fastqc/). The raw sequence reads were aligned to the human reference genome GRCh38 using the STAR aligner (ver. 2.7.11)[45]. Gene expression levels were quantified using the featureCounts tool as part of the Rsubread package (ver. 2.16.1)[46]. FeatureCounts assigns reads to genomic features, such as exons and genes, to produce count data representing the abundance of each gene in the samples. Three distinct DE analyses were performed for comparisons across the different conditions using the DESeq2 R package (ver. 1.42.1)[47]. These three conditions compared High and Low quality organoid samples according to the Expert's evaluation (Expert), k-means clustering based on five morphological parameters (Cluster_5), and Feret diameter (Feret) threshold 3050 mcm. To adjust the differences between different cell lines, the design formula included Cell lines as a covariate (design: ~Condition + Cell_lines). Lowly expressed genes were excluded using the quartile method by retaining only those with a mean expression above the first quartile (25th percentile) across all samples prior to differential gene expression analysis. For each condition (Expert, Cluster_5, Feret), differential expression analysis was performed using the DESeq2 pipeline. Only genes with an adjusted $p$-value (padj) less than 0.05 and a $|\log_2FC| > 1$ are considered significantly differentially expressed. PCA was performed on the VST-transformed data for each condition to visualize the variance in gene expression between the different experimental groups. PCA plots were generated using the ggplot2 R package (ver. 3.5.1) for aesthetic adjustments, including color-coding of conditions ('Low', 'High') using a custom color palette. ENSEMBL gene identifiers were mapped to gene symbols using the org.Hs.eg.db annotation package (ver. 3.18.0). Gene Ontology (GO) enrichment analysis was conducted using the clusterprofiler (ver. 4.10.1)[48] and enrichplot (ver. 1.22.0) (https://github.com/YuLab-SMU/enrichplot) packages. Gene Ontology enrichment analysis was performed on the significantly differentially expressed genes (padj < 0.05; $|\log_2FC| > 1$) for each condition using the clusterProfiler package (ver. 4.10.1). Venn diagram was made with eulerr (ver. 7.0.2) package. As background genes for GO enrichment analysis, we used genes included in the differential gene expression analysis. Differential gene expression and GO analysis and visualization have been performed in RStudio software (ver. 2023.09.1 + 494).

## BayesPrism deconvolution assay and WebCSEA

To estimate approximate cell type proportions in the organoids, we performed a deconvolution analysis using BayesPrism (ver. 2.0)[27]. As a reference, we used samples from organoids produced using the Lancaster protocol between day 20 and day 50 from the Human Neural Organoid Cell Atlas[28]. We used the level 3 annotation, removing labels with fewer than 100 cells in the selected samples, leaving 39898 cells with 19 labels. Protein-coding genes excluding mitochondrial, ribosomal, sex chromosome and MALAT1 genes were used to perform the deconvolution.

WebCSEA is a previously published open-access online tool to predict the cellular composition in bulk RNA-seq samples[29]. The 42 genes commonly differentially expressed in the expert evaluation, clustering and Feret diameter based discrimination of samples are available in the source data (tab 3B). The genes include non-neuronal genes such as COL8A2 and MMP9.

## Correlation analysis

We assessed the correlation between the proportion of mesenchymal cells (MC) and the Feret diameter of organoids using Pearson's correlation coefficient. We calculated the coefficient (r) and its associated $p$-value to evaluate the strength and significance of this correlation.

For analysis and visualization, we used R (version4.0.2) (R Core Team (2023)) along with the tidyverse (version2.0.0) package[49] or Python 3.0 with the seaborn plugin and statistical calculation with scipy.stats.

## K-means clustering analysis

K-means clustering was utilized to objectively categorize organoid groups based on their morphological data using R version4.0.2 (R Core Team (2023)), along with the stats, tidyverse (version2.0.0)[49], and factoextra (version1.0.7) packages. Clustering was performed using five morphological parameters (Feret diameter, Area, Cysts Amount, Cysts Area, and Perimeter), chosen after quality correlation analysis. The optimal number of clusters (k) was determined using the elbow method (Supplementary Fig. S1). The total within-cluster sum of squares (WSS) was calculated for k values ranging from 1 to 15. The 'elbow' point in the plot of WSS against k indicated the optimal number of clusters (limiting k to >1), resulting in k = 2 for both analyses. Data was normalized using Z-score normalization. K-means clustering was performed with 20 random starts to increase the likelihood of finding the global optimum. The random seed was set to 111 for reproducibility in both clustering analyses.

## Youden's J statistic

We used Youden's J statistic to evaluate nine morphological parameters (Feret diameter, Area, Cyst Amount, Cyst Area, Perimeter, Circularity, Aspect Ratio, Roundness, and Solidity) for their ability to distinguish between high quality and low quality organoids. We used R version 4.0.2 along with the stats, cutpointr (ver. 1.1.2)[50], tidyverse (ver. 2.0.0)[49], and pROC (ver. 1.18.5)[51] packages. For each parameter, the optimal cutoff point was determined using the cutpointr function. Youden's index (J) was calculated using the formula: $J = Sensitivity + Specificity - 1$. Sensitivity and specificity were computed across all cutoff values for each parameter, with the optimal cutoff identified where Youden's Index was maximized, indicating the best balance.

## Statistical analysis

Sample variance was quantified according to the coefficient of variation: $CV = \frac{SD}{mean} \times 100$. Statistical analyses were performed using RStudio (ver. 2023.09.1 + 494), Python 3.0 and Prism (ver. 10.2.0). The figures' legends include details for statistical analyses, including replicate numbers. For pair-wise comparisons, a significance was assumed when the $p$-value was below 0.05.

## Reporting summary

Further information on research design is available in the Nature Portfolio Reporting Summary linked to this article.

## Data availability

Data has been deposited at the European Genome-phenome Archive (EGA), which is hosted by the EBI and the CRG, under accession number EGAS50000000659. Further information about EGA can be found at https://ega-archive.org and "The European Genome-phenome Archive of human data consented for biomedical research. Source data is provided in Supplementary Data 1.

## Materials availability

Please note that there are restrictions to the availability of UKER iPSC lines due to the vote of the ethics committee.

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

## Acknowledgements

This study was supported by the Bavarian Ministry of Science and the Arts in the framework of the Bavarian Research Consortium 'Interaction of Human Brain Cells' (ForInter) network. Additional support came from the Else Kröner-Fresenius-Stiftung (2024_EKEA.50 (F.K.)), German Research Foundation, DFG (WI 3567/2-1 (B.W.); GRK2162/ 270949263 (B.W. and J.W.), CRU5024/ 505539112 WI 3567/4-1 (B.W.) and WI 1620/4-1 (J.W.)), the TreatHSP consortium (BMBF 01GM1905B, 01GM2209B to B.W., M.R. and J.W.) and the Interdisziplinäres Zentrum für Klinische Forschung (IZKF) (Erstantragsteller project J-112 (D.K.)); ELAN-Fond P153 (F.K.). The authors thank Holger Wend for excellent technical support. The present work was performed in (partial) fulfillment of the requirements for obtaining the degree 'Dr. med.' (T.B.). Graphical illustrations were made with BioRender.com.

## Author contributions

Conceptualization: D.K., T.B., B.W., F.K., M.K. and S.F.; Methodology: T.B., D.K., M.F., S.P., L.Z., S.F., S.P., M.F., P.G. and E.G.; Investigation: T.B., D.K., M.F., N.Z., F.F., N.N., M.B. and E.G.; Formal analysis: T.B., D.K., N.Z., and F.K.; Visualization: T.B., D.K., N.Z. and L.Z.; Data curation: T.B., D.K., and N.Z.; Original draft preparation: D.K., F.K. and B.W.; Writing—review and editing: T.B., D.K., F.K., N.Z., B.W. and F.T.; Resources: F.K., B.W., M.K., J.W. and I.P.; Project administration: D.K., F.K. and B.W.; Funding acquisition: B.W., M.R., J.W., C.G., D.K. and F.K. All authors critically revised the manuscript and approved the submitted version.

## Funding

## Competing interests

The authors declare no competing interests.

## Ethics statement

The human cells used in this study were handled in accordance with the principles outlined in the Declaration of Helsinki. Concerning all UKER lines, informed written consent was obtained from the participating individuals to use donor tissue for research purposes. The generation and use of local human iPSC lines were approved by the Institutional Review Board of the University Hospital of Erlangen (Nr. 4120 and 259_17B: Generation of

human neuronal models for neurodegenerative diseases). Concerning iPSC lines Kucg2 and Sojd3 (HipSci feeder-free panel (ECACC 77659901)), the MTA was obtained from the Wellcome Trust Sanger Institute for research purposes. The remaining iPSC lines (TMOi001A, KOLF2.1, CVB) are commercially available. The use of human embryonic stem cells for this project was approved by the Central Ethics Committee for Stem Cell Research (121. Approval according to the German Stem Cell Act to Beate Winner).
