## [Transparent Peer Review file · Communications Biology]

Deciphering Brain Organoids Heterogeneity by Identifying Key Quality Determinants

Corresponding Author: Dr Florian Krach

Version 0:

Reviewer comments:

Reviewer #1

(Remarks to the Author)

In the manuscript entitled “Deciphering Brain Organoids Heterogeneity by Identifying Key Quality Determinants”, Kachkin et al. describe a morphological evaluation of brain organoids from 12 independent, healthy donor human pluripotent stem cell (hPSC) lines at differentiation day 30 to identify features that correlate with “organoid quality” as defined by an expert evaluator. Defining objective criteria that can accurately predict the quality of hPSC-derived organoids early in the differentiation process addresses the problem of heterogeneity in brain organoid cultures, a known and significant impediment to the use hPSC-derived brain organoids. The manuscript is well-laid out, easy to follow, and describes work that can be expanded upon in subsequent experiments. A strength of the work is the use of 12 healthy donor hPSC lines in these experiments to capture a representative sample of hPSC organoid potential. There were a few concerns that diminished enthusiasm for the manuscript.

Major Comments

1. On page 6 it is noted that “an expert with more than 5 years of experience visually evaluated and rated the organoids’ quality as either ‘good’ or ‘bad’”. There is no further description of the qualifications of the expert, what is meant by either ‘good’ or ‘bad’ in terms of organoid generation, the characteristics the expert was evaluating, the correct predictive rate of the expert (i.e., how many ‘good’ organoids at day 30 were actually ‘good’ organoids at some later timepoint when used for an experiment) or whether the expert was blinded to the hPSC line from which each individual organoid came from. This raises concerns about the overall evaluation as the objective measures are compared back to the expert’s opinion to determine whether they correlate and are relevant organoid outcome. It would be helpful to clarify information about the parameters associated with the expert opinion as noted above. In an ideal scenario, it may have been better to state the characteristics of a ‘good’ organoid (preferably something measurable such as 3 or more ventricular-like structures or electrically active, etc.) and hypothesize that the 9 parameters measured characteristics, when measured at day 30, correlated with the stated ‘good’ outcome at the point at which the organoids would be used thus removing the need for an expert opinion.
2. Sample sizes are stated throughout the manuscript (for example, 6 organoids per line for immunostaining analysis, 6 organoids per line evaluated by the expert, etc.). It is not clear how the authors determined that each sample size was appropriate for the evaluation being done. This question is important as the premise of the manuscript is that organoids are highly variable. A consultation with biostatistician could be useful here.
3. In Figure 3G, COL1A1 immunostaining is used to confirm the presence of mesenchymal cells. As generic fibroblasts can also express COL1A1, it would be useful to have additional immunostaining to support the statement that mesenchymal cells are truly the cell source responsible for ‘bad’ organoids. This is relevant because the GO analysis suggests a broader cell source than what is annotated in the Human Neural Organoid Cell Atlas (HNOCA). Deconvolution algorithms can assign cells to a poorly matched category because it is the “best” match out of even more poorly aligned categories.
4. It would be helpful to include a table in the body of the manuscript that describes what the 9 parameters are that are being measured and what the measurement is. While I appreciate the artistic imagery of Figure 2A, the font size is small, and it is not clear what the actual measurements are. For example, how is ‘circularity’ different than ‘roundness’? The 9 parameters measured are noted in the Methods and Materials (p13) and Table S2; however, no description is provided. At a minimum, if the definitions of the measurements are consistent with the definitions used in ImageJ, this should be noted to aid the reader in interpreting the characteristics.

Minor Comments

1. In the abstract (p3), the term “organdie” is used. This is a non-standard term. It seems to be used synonymously with “organoid”. Please use the term “organoid” throughout for better clarity.
2. It is unclear what information the statement on page 3 “Recent discoveries in molecular and cellular anthropology and human brain evolution underlie their methodological power” is trying to convey relevant to the topic of the manuscript. Consider clarifying the statement and its relationship to the theme of the manuscript or removing.
3. It would be helpful to provide a list of the 42 differentially expressed genes discussed in Figure 3B.

Reviewer #2

(Remarks to the Author)

In the manuscript titled "Deciphering Brain Organoids Heterogeneity by Identifying Key Quality Determinants" by Kachkin et al., the authors generate and analyze neural organoids from 12 distinct lines, identifying key factors that influence organoid quality, including morphological parameters and cellular composition. While the authors provide a comprehensive and rigorous analysis, and their findings hold promise for advancing quality control in the neural organoid field, there are areas where the manuscript could be strengthened further. See specific comments below.

Specific Comments:

-Figure 1 (C-F): The authors present data showing variability in organoid cellular composition using antibody detection by immunofluorescence and flow cytometry. While it is understandable that there would be detection differences between the 2 techniques there is a substantial discrepancy in the measurements for Map2 positive cells based on the image shown in Fig1C. Based on the image on Fig1C one would expect that HUES6-derived organoids would have significantly more Map2 positive cells compared to CVB-derived organoids (that shows no Map2 positive cells in the image). The Flow cytometry results show the opposite (Fig 1 F), with CVB organoids having many more Map2 positive cells. How reliable are the measurements? There is no statistical analysis on Fig1, so it is difficult to interpret the meaning of the data displayed.

-The authors should provide a schematic showing the actual Ferret diameter measurement used on a “good” or “bad” organoid image as an illustration example.

-The authors should test possible alternatives to improve organoid quality and decrease variability in cellular composition and morphology. Ex: In organoids with increased mesenchymal cell composition, would it be useful to propose addition of a small molecule to inhibit mesenchymal cell fate?

-How does the different cell populations observed by the authors in the different organoids using the deconvolution method on bulk RNA correlates to single cell data? It would be important to at least know how well the deconvolution data correlates to the actual single cell data using the authors' methodology.

-Figure 3D: The percentage of “good” and “bad” organoids is not evenly distributed between the different cell lines analyzed, with cell lines with 100% “good” organoids (HuES6, TMOi001A, Sojd3, 33Q, H9) and cell lines with 100% “bad” organoids (CVB, RN4, KOLF2.1J). Have the authors controlled for genetic background as a confounding factor in the comparative analysis? This is critical because genetic background will affect the expression profiles.

Minor Comment:

The word "organoid" is misspelled as "organdie" a few times in the abstract

Reviewer #3

(Remarks to the Author)

In this study, the authors discuss the experimental variability of brain organoids produced from human pluripotent stem cells and the need for organoid selection criteria for analysis. Using 72 brain organoids from different hPSC lines, the authors identified ferret diameter through correlation analysis of morphological or cytological characteristics with brain organoid quality. The smaller the ferret diameter, the higher the quality of the brain organoids.

The authors observed a higher percentage of mesenchymal cells in lower quality brain organoids by transcriptome analysis and observed a positive correlation between the percentage of mesenchymal cells and ferret diameter. This study suggests that ferret diameter can serve as a surrogate marker for high-quality organoids.

While this study is concise and provides compelling evidence for a role of ferret diameter in high-quality brain organoids, it falls short of providing mechanistic understanding or significant biological insight into how mesenchymal cells function and inhibit brain organoid function.

A deeper mechanistic understanding of how mesenchymal cells influence brain organoid differentiation will clarify this aspect. The authors may also establish a framework for the stable production of high-quality brain organoids by controlling the expression of mesenchymal cells.

I have some concerns, as detailed below.

Major comments

1) The authors relied heavily on GO analysis to confirm mesenchymal cells (as depicted in Figure 3). As a mesenchymal cell-associated protein, the results are validated by immunocytochemical (ICC) analysis of COL1A1, but this is a

comparative validation only in two lines, and COL1A1 expression and ferret diameter should be compared in the 12 hPSC lines used in this study.

Minor comments

- 1) Fig. 1B. Please make it a little brighter.
- 2) Fig. 1D, E, F. It is preferable to describe the vertical and horizontal axes.
- 3) Fig. 3H. There is no mention of Fig. 3H in the text.
- 4) Fig. 3F. The correct value is "r=0.59." "r=-0.59" is incorrect. A significant negative correlation" is indicated in the text, but this should be corrected since there is a positive correlation.
- 5) Fig. 3S. I felt there was not enough explanation in the text; please add an explanation of WebCSEA, an explanation of the 42 genes, and an explanation of their relevance to non-neuronal cells.
- 6) Fig. 1S, 2S, 3S. There is an error in the Fig number of the Legend title.

Version 1:

Reviewer comments:

Reviewer #1

(Remarks to the Author)

In the revised manuscript by Boerstler et al., the authors describe an unbiased approach for determining the quality of brain organoids. As the use of brain organoids increases, clearly defined and standardized approaches for assessing brain organoid quality will continue to increase in importance. The authors address key concerns raised during the initial review of the manuscript. Specifically, the authors better describe their procedures for evaluating the quality of organoids and provide additional data supporting the correlation between mesenchymal cell numbers and an organoid quality. Revisions to figures and the text more clearly articulate the authors' work and interpretation of their data. Similarly, additions to the supplementary data and information assist the reader in better understanding and evaluating the work.

While other published articles investigate organoid quality, the manuscript contributes relevant information to the field and adds to the weight of evidence for using organoid size and cellular context to improve quality and consistency of organoid production.

Sufficient detail is provided in the Methods and Materials sections for those working in the field to evaluate and replicate the work as needed to validate and extend the authors' work.

One minor comment is that, in the graphical abstract, the word "transcriptome" is missing the terminal "e".

Reviewer #2

(Remarks to the Author)

The authors have properly addressed all my comments. I have no further comments at this point.

Dear Dr. Bessieres

We are pleased to submit the revised version of our Manuscript entitled “Deciphering Brain Organoids Heterogeneity by Identifying Key Quality Determinants” (COMMSBIO-24-8692). We thank you and the reviewers for the kind and positive responses. The reviewers brought up questions, which we carefully answered and included, according to their suggestions. We briefly address the major changes here, followed by a detailed point-by-point response below.

The major request of reviewer 1 and 2 was to clarify and extend information about the parameters associated with the quality rating of the organoid. We now added more detailed information of this process based on the criteria defined in the original publication of the generation of unguided cerebral organoids (*Lancaster et al., 2014; PMID: 25188634*). In addition, we now include schematic information about the morphological analysis.

Another request, which both reviewer 1 and 3 pointed to, was to expand the analysis of mesodermal markers throughout all cell lines. In this respect, we now analyzed the mesodermal markers CD73 and CD105 and performed correlation analyses now included in Figure 4.

In summary, we are pleased that the new experiments, re-analyses, and clarifications have significantly strengthened and extended the conclusions of the original manuscript. We very much hope that our revised manuscript will be acceptable for publication in *Communications Biology*.

Yours sincerely,

Florian Krach, MD PhD on behalf of all authors

Reviewer #1 (Remarks to the Author):

In the manuscript entitled “Deciphering Brain Organoids Heterogeneity by Identifying Key Quality Determinants”, Kachkin et al. describe a morphological evaluation of brain organoids from 12 independent, healthy donor human pluripotent stem cell (hPSC) lines at differentiation day 30 to identify features that correlate with “organoid quality” as defined by an expert evaluator. Defining objective criteria that can accurately predict the quality of hPSC-derived organoids early in the differentiation process addresses the problem of heterogeneity in brain organoid cultures, a known and significant impediment to the use hPSC-derived brain organoids. The manuscript is well-laid out, easy to follow, and describes work that can be expanded upon in subsequent experiments. A strength of the work is the use of 12 healthy donor hPSC lines in these experiments to capture a representative sample of hPSC organoid potential. There were a few concerns that diminished enthusiasm for the manuscript.

Major Comments

1. On page 6 it is noted that “an expert with more than 5 years of experience visually evaluated and rated the organoids’ quality as either ‘good’ or ‘bad’”. There is no further description of the qualifications of the expert, what is meant by either ‘good’ or ‘bad’ in terms of organoid generation, the characteristics the expert was evaluating, the correct predictive rate of the expert (i.e., how many ‘good’ organoids at day 30 were actually ‘good’ organoids at a later timepoint when used for an experiment) or whether the expert was blinded to the hPSC line from which each individual organoid came from. This raises concerns about the overall evaluation as the objective measures are compared back to the expert’s opinion to determine whether they correlate and are relevant organoid outcome. It would be helpful to clarify information about the parameters associated with the expert opinion as noted above. In an ideal scenario, it may have been better to state the characteristics of a ‘good’ organoid (preferably something measurable such as 3 or more ventricular-like structures or electrically active, etc.) and hypothesize that the 9 parameters measured characteristics, when measured at day 30, correlated with the stated ‘good’ outcome at the point at which the organoids would be used thus removing the need for an expert opinion.

Response:

The reviewer’s comments relate to the objectivity and criteria of the expert’s quality evaluation. We thank the reviewer for highlighting this critical problem, which we are addressing in our study to identify quantitative metrics for organoid quality that are currently limited. Quality evaluation of organoids in the field is presently based on qualitative, not quantitative, assessment. We used criteria from the original publication of the generation of unguided cerebral organoids 10.1038/nprot.2014.158. High quality (previous “Good”) cerebral organoids at day 30 were defined by the presence of neuroepithelial bulbs that had migrated into the Matrigel embedding and tissue exhibiting regular shape. Low quality (previous “Bad”) non-cerebral organoids were

classified when they exhibited overt, large, fluid-filled cyst-like structures, irregularly outgrowing cells, or irregular tissue structure. We added to our manuscript the missing information:

Following, cerebral organoids were classified as high-quality according to the critical morphological hallmark of a spherical shape interrupted by neuroepithelial buds growing into the Matrigel embedding (Figure 1B, green). Low-quality organoids were classified by the presence of overt large fluid-filled cysts, overt migrating cells, or an irregular shape (Figure 1B, red). Ref: Lancaster et al., 2014; PMID: 25188634, Velasco et al., 2019 PMID: 31168097

2. Sample sizes are stated throughout the manuscript (for example, 6 organoids per line for immunostaining analysis, 6 organoids per line evaluated by the expert, etc.). It is not clear how the authors determined that each sample size was appropriate for the evaluation being done. This question is important as the premise of the manuscript is that organoids are highly variable. A consultation with biostatistician could be useful here.

Response:

We agree with the reviewer about the importance of an appropriate sample size. We did not use statistical methods to pre-determine the sample size. Sample sizes in the field are usually determined empirically. We used a similar sample size to that of other previous publications. (Ref.:Amin et al., 2024; PMID: 39642864, Zenkt et al., 2024 PMID: 38914828, Accurate biostatistical sample size calculations require knowledge about variability and variance, however, quantitative knowledge in a larger size of PSC lines is limited. To address this interline variability, we used a large cohort comprising hPSC lines from 12 different donors.

3. In Figure 3G, COL1A1 immunostaining is used to confirm the presence of mesenchymal cells. As generic fibroblasts can also express COL1A1, it would be useful to have additional immunostaining to support the statement that mesenchymal cells are truly the cell source responsible for 'bad' organoids. This is relevant because the GO analysis suggests a broader cell source than what is annotated in the Human Neural organoid Cell Atlas (HNOCA). Deconvolution algorithms can assign cells to a poorly matched category because it is the "best" match out of even more poorly aligned categories.

Response:

We agree with the reviewer's suggestion that COL1A1 may be generic and have instead analyzed two independent mesenchymal cell-associated markers, CD73 and CD105, in all 12 PSC lines. We identify a positive correlation of both markers with each other, giving us confidence that the markers measure cells of similar identity. We identify for both markers a significant correlation with the organoid section's Feret diameter (see new Figure 4). PSC lines that in their majority produce high quality organoids exhibit lower CD105 and CD73

fluorescent intensities (see new Supplementary Figure S6). We also performed a new correlation of the gene expression of the two markers ENG (encoding CD105) and NT5E (encoding CD73) with the Feret diameter and predicted mesenchymal cell content. We identify a significantly positive correlation for all mesenchymal cell-associated genes, supporting our finding (Supplementary Figure S5). We hope by employing two independent markers, we are able to convince the reviewer of the results obtained by the deconvolution algorithm.

Figure 4. Experimental validation illustrating that mesenchymal cells are associated with higher organoid Feret diameter in sections A. Representative images of organoids from four hPSC lines. Stained with CD73 (blue), CD105 (yellow), MAP2 (magenta), and DAPI. Scale bar 500 μ m. **B** Scatter plot depicting mean fluorescent intensity of CD105 (x-axis) and CD73 (y-axis). Individual dots represent individual organoids ($n=65$). Color depicts 12 PSC lines (legend on right). Significant positive correlation (Spearman $r=0.7$, $p=1.08 \times 10^{-10}$) is depicted in the graph. **C** Scatter plot depicting the organoid section's Feret diameter (x-axis) and mean fluorescent intensity of CD105 (y-axis). Individual dots represent individual organoids ($n=65$). Color depicts 12 PSC lines (legend on right). Significant positive correlation (Spearman $r=0.44$, $p=0.00029$) is depicted in the graph. **D** Scatter plot depicting the organoid section's Feret diameter (x-axis) and mean fluorescent intensity of CD73 (y-axis). Individual dots represent individual organoids ($n=65$). Color depicts 12 PSC lines (legend on right). Significant positive correlation (Spearman $r=0.39$, $p=0.00146$) is depicted in the graph.

Supplementary Figure S6. CD105 and CD73 intensity in organoids based on PSC-line organoid majority quality rating

Supplementary Figure S6. CD105 and CD73 intensity in organoids based on PSC-line organoid majority quality rating. The 12 PSC lines used in the study were separated in two groups (majority as high or low quality). Fluorescence intensity of CD105 (left) or CD73 (right) was quantified. Graphs are box plots with median as central line. P-values depicted on top (Rank-sum test). Individual organoids depicted as dots.

Supplementary Figure S5. Correlation of Feret diameter and predicted mesenchymal cell content with gene expression of ENG and NT5E

Supplementary Figure S5. Correlation of Feret diameter and predicted mesenchymal cell content with gene expression of ENG and NT5E. Scatter plots correlating Feret diameter (left plots) or predicted mesenchymal cell content (right) from RNA-seq deconvolution with gene expression (DESeq2 normalized counts) of mesenchymal cell associated genes ENG (CD105), NT5E (CD73). Individual organoid classification by expert depicted in red dots (LOW) and green dots (HIGH). Correlation coefficient (Pearson and Spearman) and regression coefficient with respective P values on top of each graph.

4. It would be helpful to include a table in the body of the manuscript that describes what the 9 parameters are that are being measured and what the measurement is. While I appreciate the artistic imagery of Figure 2A, the font size is small, and it is not clear what the actual measurements are. For example, how is 'circularity' different than 'roundness'? The 9 parameters measured are noted in the Methods and Materials (p13) and Table S2; however, no description is provided. At a minimum, if the definitions of the measurements are consistent with the definitions used in ImageJ, this should be noted to aid the reader in interpreting the characteristics.

Response: We thank the reviewer for this suggestion and agree this would be helpful for the reader for interpretation of the results. We added a column to Table S2 with a description of the individual parameter.

Minor Comments

1. In the abstract (p3), the term "organdie" is used. This is a non-standard term. It seems to be used synonymously with "organoid". Please use the term "organoid" throughout for better clarity.

Response: We thank the reviewer for their careful reading and confirm we have changed this typo in the manuscript.

2. It is unclear what information the statement on page 3 "Recent discoveries in molecular and cellular anthropology and human brain evolution underlie their methodological power" is trying to convey relevant to the topic of the manuscript. Consider clarifying the statement and its relationship to the theme of the manuscript or removing.

Response: We thank the reviewer for their feedback and have taken out this sentence.

3. It would be helpful to provide a list of the 42 differentially expressed genes discussed in Figure 3B.

Response: We agree with the reviewer that this is interesting information. The information is available for readers in the source data (Tab '3B')

Reviewer #2 (Remarks to the Author):

In the manuscript titled "Deciphering Brain Organoids Heterogeneity by Identifying Key Quality Determinants" by Kachkin et al., the authors generate and analyze neural organoids from 12 distinct lines, identifying key factors that influence organoid quality, including morphological parameters and cellular composition. While the authors provide a comprehensive and rigorous analysis, and their findings hold promise for advancing quality control in the neural organoid field, there are areas where the manuscript could be strengthened further. See specific comments below.

Specific Comments:

-Figure 1 (C-F): The authors present data showing variability in organoid cellular composition using antibody detection by immunofluorescence and flow cytometry. While it is understandable that there would be detection differences between the 2 techniques there is a substantial discrepancy in the measurements for Map2 positive cells based on the image shown in Fig1C. Based on the image on Fig1C one would expect that HUES6-derived organoids would have significantly more Map2 positive cells compared to CVB-derived organoids (that shows no Map2 positive cells in the image). The Flow cytometry results show the opposite (Fig 1 F), with CVB organoids having many more Map2 positive cells. How reliable are the measurements? There is no statistical analysis on Fig1, so it is difficult to interpret the meaning of the data displayed.

Response:

We appreciate the reviewer's careful reading and critical questions. Indeed, the results from MAP2 IF imaging and flow cytometry indicate high variability within the cell line group. Another explanation is that flow cytometry quantifies single-cell data, whereas the IF image shows the overt distribution of long MAP2 neurites, making it difficult to compare in numbers. We agree with the reviewers that a single-cell quantification would be preferable. However, to our knowledge, there is no robust single-cell analysis of IF images from neurite markers from brain organoid sections.

-The authors should provide a schematic showing the actual Ferret diameter measurement used on a "good" or "bad" organoid image as an illustration example.

Response: We thank the reviewer for this suggestion to clarify the metric better for the reader. We added a supplementary figure to illustrate 4 representative organoids used for the analysis and to schematically illustrate the Ferret diameter. (see Fig. S3)

Supplementary Figure S3. Schematic representation of Feret diameter

Supplementary Figure S3. Schematic representation of Feret diameter
4 representative organoid images used for analysis of high-quality and low-quality organoids in 4 different hPSC-lines. Schematic Feret diameter as red arrow. Low-quality organoids exhibit a tendency for a higher Feret diameter.

-The authors should test possible alternatives to improve organoid quality and decrease variability in cellular composition and morphology. Ex: In organoids with increased mesenchymal cell composition, would it be useful to propose addition of a small molecule to inhibit mesenchymal cell fate?

Response:

We appreciate the reviewer's suggestion using small molecules to inhibit mesenchymal cell development during organoid differentiation to validate our results. We used the original version of the unguided cerebral organoid differentiation protocol with a minimum of morphogens to investigate reliable markers identifying the differentiation variability (Lancaster et. al, 2014; PMID: 25188634) .

However, we added in the discussion that this is a great future direction and outlined possible pathways that could be modulated to reduce mesenchymal cell content.

Other protocols already use specific small-molecule inhibitors targeting the TGF- β , BMP pathways. Molecules such as SB431542 and A83-01 are used during neural induction, where the addition of SB431542 skews iPSCs toward neuroectoderm by blocking mesodermal signals (Song et al., 2021; PMID: 34234646, Wu et al., 2023; PMID: 37178118). Additionally,

the BMP pathways can be inhibited during neural induction to inhibit mesodermal differentiation. Compounds such as Noggin (a BMP antagonist) or small molecules like Dorsomorphin and LDN-193189 (inhibitors of BMP type I receptors) are used to inhibit BMP-driven mesodermal and epidermal differentiation. Amin et al., 2024; PMID: 39642864. Our manuscript indicates mesodermal differentiation as one of the major confounders in neuroectodermal differentiation and highlights the need for further compound screening to increase reliability in cerebral organoid differentiation. We agree that this mechanistic study would be a logical next step to promote more robust organoid differentiation. However, we believe this may be out of the scope of this manuscript.

-How does the different cell populations observed by the authors in the different organoids using the deconvolution method on bulk RNA correlates to single cell data? It would be important to at least know how well the deconvolution data correlates to the actual single cell data using the authors' methodology.

Response: We agree with the reviewer that this would be interesting to compare to promote the computational approach. However, we do not have corresponding single-cell RNA-seq data of those exact samples available. Additionally, a true comparability study is not feasible as it is not possible to use the exact same sample for scRNA-seq and bulk RNA-seq. To set our findings in the context to illustrate the suitability of the approach: Mesenchymal cells were detected using scRNA-seq in an early study looking at unguided cerebral organoids from a single iPSC line and their presence was validated using COL1A1 staining. However, no exact number of how many percent of mesenchymal cells are present is given (Camp et al., 2015; PMID: 26644564). Another publication investigating cellular composition via scRNA-seq in 31 cerebral organoids that are 3 or 6 months old in a single iPSC background found presence of a cell population they annotated as mesodermal cells (3029 of 82,291 sequenced cells over all samples = 3.7%). The mesenchymal cell content in our study ranges from 0.5%-74.1%. In regard to mesenchymal cell content in our study, we now provide experimental data from all 12 PSC lines. We identify a positive correlation of CD73 and CD105, two mesenchymal cell-associated markers, with each other, giving us confidence that the markers measure cells of similar identity. We identify a significant correlation between both markers and the Feret diameter of the organoid section (see new Figure 4). PSC lines that, in their majority produce high quality organoids exhibit lower CD105 and CD73 fluorescent intensities (see new Supplementary Figure S6). We also performed a new correlation of the gene expression of the two markers ENG (encoding CD105) and NT5E (encoding CD73) with the Feret diameter and predicted mesenchymal cell content. We identify a significantly positive correlation for all mesenchymal cell-associated genes, supporting our finding (Supplementary Figure S5). We hope by employing two independent markers, we are able to convince the reviewer of the

results obtained by the deconvolution algorithm. We hope to address the reviewer's concerns with this new data.

Figure 4. Experimental validation illustrating that mesenchymal cells are associated with higher organoid Feret diameter in sections A. Representative images of organoids from four hPSC lines. Stained with CD73 (blue), CD105 (yellow), MAP2 (magenta), and DAPI. Scale bar 500 μm . **B** Scatter plot depicting mean fluorescent intensity of CD105 (x-axis) and CD73 (y-axis). Individual dots represent individual organoids ($n=65$). Color depicts 12 PSC lines (legend on right). Significant positive correlation (Spearman $r=0.7$, $p=1.08 \times 10^{-10}$) is depicted in the graph. **C** Scatter plot depicting the organoid section's Feret diameter (x-axis) and mean fluorescent intensity of CD105 (y-axis). Individual dots represent individual organoids ($n=65$). Color depicts 12 PSC lines (legend on right). Significant positive correlation (Spearman $r=0.44$, $p=0.00029$) is depicted in the graph. **D** Scatter plot depicting the organoid section's Feret diameter (x-axis) and mean fluorescent intensity of CD73 (y-axis). Individual dots represent individual organoids ($n=65$). Color depicts 12 PSC lines (legend on right). Significant positive correlation (Spearman $r=0.39$, $p=0.00146$) is depicted in the graph.

Supplementary Figure S6. CD105 and CD73 intensity in organoids based on PSC-line organoid majority quality rating

Supplementary Figure S6. CD105 and CD73 intensity in organoids based on PSC-line organoid majority quality rating. The 12 PSC lines used in the study were separated in two groups (majority as high or low quality). Fluorescence intensity of CD105 (left) or CD73 (right) was quantified. Graphs are box plots with median as central line. P-values depicted on top (Rank-sum test). Individual organoids depicted as dots.

Supplementary Figure S5. Correlation of Feret diameter and predicted mesenchymal cell content with gene expression of ENG and NT5E

Supplementary Figure S5. Correlation of Feret diameter and predicted mesenchymal cell content with gene expression of ENG and NT5E. Scatter plots correlating Feret diameter (left plots) or predicted mesenchymal cell content (right) from RNA-seq deconvolution with gene expression (DESeq2 normalized counts) of mesenchymal cell associated genes ENG (CD105), NT5E (CD73). Individual organoid classification by expert depicted in red dots (LOW) and green dots (HIGH). Correlation coefficient (Pearson and Spearman) and regression coefficient with respective P values on top of each graph.

-Figure 3D: The percentage of “good” and “bad” organoids is not evenly distributed between the different cell lines analyzed, with cell lines with 100% “good” organoids (HuES6, TMOi001A, Sojd3, 33Q, H9) and cell lines with 100% “bad” organoids (CVB, RN4, KOLF2.1J). Have the authors controlled for genetic background as a confounding factor in the comparative analysis? This is critical because genetic background will affect the expression profiles.

Response: We agree with the reviewer's observation that some PSC lines exhibit a better quality profile than others, and careful consideration of confounding factors is necessary. We confirm that the genetic background was controlled for in our gene expression analyses. This information is available in the methods section.

Minor Comment:

The word "organoid" is misspelled as "organdie" a few times in the abstract

Response: We thank the reviewer for their careful reading and confirm we have changed this typo in the manuscript.

Reviewer #3 (Remarks to the Author):

In this study, the authors discuss the experimental variability of brain organoids produced from human pluripotent stem cells and the need for organoid selection criteria for analysis. Using 72 brain organoids from different hPSC lines, the authors identified ferret diameter through correlation analysis of morphological or cytological characteristics with brain organoid quality. The smaller the ferret diameter, the higher the quality of the brain organoids.

The authors observed a higher percentage of mesenchymal cells in lower quality brain organoids by transcriptome analysis and observed a positive correlation between the percentage of mesenchymal cells and ferret diameter. This study suggests that ferret diameter can serve as a surrogate marker for high-quality organoids.

While this study is concise and provides compelling evidence for a role of ferret diameter in high-quality brain organoids, it falls short of providing mechanistic understanding or significant biological insight into how mesenchymal cells function and inhibit brain organoid function.

A deeper mechanistic understanding of how mesenchymal cells influence brain organoid differentiation will clarify this aspect. The authors may also establish a framework for the stable production of high-quality brain organoids by controlling the expression of mesenchymal cells.

I have some concerns, as detailed below.

Major comments

- 1) The authors relied heavily on GO analysis to confirm mesenchymal cells (as depicted in Figure 3). As a mesenchymal cell-associated protein, the results are validated by immunocytochemical (ICC) analysis of COL1A1, but this is a comparative validation

only in two lines, and COL1A1 expression and ferret diameter should be compared in the 12 hPSC lines used in this study.

Response: We acknowledge the reviewer's concern and performed staining of two mesenchymal cell-associated markers in all 12 organoids and also measured the Feret diameter of the respective sections. As pointed out by Reviewer 2, COL1A1 may be generic. Hence, we stained for CD73 and CD105, two markers that are rather associated with mesenchymal cells. We identify a positive correlation of both markers with each other, giving us confidence that the markers measure cells of similar identity. We identify for both markers a significant correlation with the Feret diameter (see new Figure 4). PSC lines that in their majority produce high quality organoids exhibit lower CD105 and CD73 fluorescent intensities (see new Supplementary Figure S6). We also performed a new correlation of the gene expression of the three markers ENG (encoding CD105) and NT5E (encoding CD73) with the Feret diameter and predicted mesenchymal cell content. We identify a significantly positive correlation for all mesenchymal cell-associated genes, supporting our finding (Supplementary figure S5). We hope that we convinced the reviewer that this confirms our computational approach.

Figure 4. Experimental validation illustrating that mesenchymal cells are associated with higher organoid Feret diameter in sections A. Representative images of organoids from four hPSC lines. Stained with CD73 (blue), CD105 (yellow), MAP2 (magenta), and DAPI. Scale bar 500 μ m. **B** Scatter plot depicting mean fluorescent intensity of CD105 (x-axis) and CD73 (y-axis). Individual dots represent individual organoids ($n=65$). Color depicts 12 PSC lines (legend on right). Significant positive correlation (Spearman $r=0.7$, $p=1.08 \times 10^{-10}$) is depicted in the graph. **C** Scatter plot depicting the organoid section's Feret diameter (x-axis) and mean fluorescent intensity of CD105 (y-axis). Individual dots represent individual organoids ($n=65$). Color depicts 12 PSC lines (legend on right). Significant positive correlation (Spearman $r=0.44$, $p=0.00029$) is depicted in the graph. **D** Scatter plot depicting the organoid section's Feret diameter (x-axis) and mean fluorescent intensity of CD73 (y-axis). Individual dots represent individual organoids ($n=65$). Color depicts 12 PSC lines (legend on right). Significant positive correlation (Spearman $r=0.39$, $p=0.00146$) is depicted in the graph.

Supplementary Figure S6. CD105 and CD73 intensity in organoids based on PSC-line organoid majority quality rating

Supplementary Figure S6. CD105 and CD73 intensity in organoids based on PSC-line organoid majority quality rating. The 12 PSC lines used in the study were separated in two groups (majority as high or low quality). Fluorescence intensity of CD105 (left) or CD73 (right) was quantified. Graphs are box plots with median as central line. P-values depicted on top (Rank-sum test). Individual organoids depicted as dots.

Supplementary Figure S5. Correlation of Feret diameter and predicted mesenchymal cell content with gene expression of ENG and NT5E

Supplementary Figure S5. Correlation of Feret diameter and predicted mesenchymal cell content with gene expression of ENG and NT5E. Scatter plots correlating Feret diameter (left plots) or predicted mesenchymal cell content (right) from RNA-seq deconvolution with gene expression (DESeq2 normalized counts) of mesenchymal cell associated genes ENG (CD105), NT5E (CD73). Individual organoid classification by expert depicted in red dots (LOW) and green dots (HIGH). Correlation coefficient (Pearson and Spearman) and regression coefficient with respective P values on top of each graph.

Minor comments

1) Fig. 1B. Please make it a little brighter.

Response:

We appreciate the reviewer's comment and have adapted the figure.

2) Fig. 1D, E, F. It is preferable to describe the vertical and horizontal axes.

Response: Fig. 1D, E and F have vertical as well as horizontal axes labels. We also checked all graphs in the manuscript and confirm that all axes are labeled.

3) Fig. 3H. There is no mention of Fig. 3H in the text.

Response:

We thank the reviewer for their careful reading, and we corrected this error.

4) Fig. 3F. The correct value is "r=0.59." "r=-0.59" is incorrect. A significant negative correlation is indicated in the text, but this should be corrected since there is a positive correlation.

Response:

We thank the reviewer for their careful reading, and we corrected this error.

5) Fig. 3S. I felt there was not enough explanation in the text; please add an explanation of WebCSEA, an explanation of the 42 genes, and an explanation of their relevance to non-neuronal cells.

Response: We further clarified WebCSEA in the results and methods section as followed and hope the information is sufficient to address the reviewer's concern:

'We validated this finding computationally by applying WebCSEA - Web-based Cell-type Specific Enrichment Analysis of Genes as a second prediction approach with an alternative underlying statistical method as BayesPrism (t statistics vs. Bayesian approach).'

'WebCSEA is a previously published open-access online tool to predict the cellular composition in bulk RNA-seq samples²⁸. The 42 genes commonly differentially expressed in the expert evaluation, clustering and Feret diameter based discrimination of samples are available in the source data (tab 3B). The genes include non-neuronal genes such as COL8A2 and MMP9.'

6) Fig. 1S, 2S, 3S. There is an error in the Fig number of the Legend title.

Response: We thank the reviewer for their careful reading, and we corrected this error.